# Untangling Effect and Side Effect: Consistent Causal Inference in Non-Targeted Trials

## Abstract

A treatment is usually appropriate for some group (the "sick" group) on whom it has an effect, but it can also have a side-effect when given to subjects from another group (the "healthy" group). In a non-targeted trial both sick and healthy subjects may be treated, producing heterogeneous effects within the treated group. Inferring the correct treatment effect on the sick population is then difficult, because the effect and side-effect are tangled. We propose an efficient nonparametric approach to untangling the effect and side-effect, called **PCM** (pre-cluster and merge). We prove its asymptotic consistency in a general setting and show, on synthetic data, more than a 10x improvement in accuracy over existing state-of-the-art.

## 1 Introduction

A standard approach to causal effect estimation is the targeted randomized controlled trial (RCT), see (8; 13; 15; 17; 23). To test a treatment's effect on a sick population, subjects are recruited and admitted into the trial based on eligibility criteria designed to identify sick subjects. The trial subjects are then randomly split into a treated group that receives the treatment and a control group that receives the best alternative treatment (or a placebo). "Targeted" means only sick individuals are admitted into the trial via the eligibility criteria, with the implicit assumption that only a single treatment-effect is to be estimated. This ignores the possibility of treated subgroups among the sick population with heterogeneous effects. Further, one often does not have the luxury of a targeted RCT. For example, eligibility criteria for admittance to the trial may not unambiguously identify sick subjects, or one may not be able to control who gets into the trial. When the treatment is not exclusively applied on sick subjects, we say the trial is non-targeted and new methods are needed to extract the treatment effect on the sick, (25). Non-targeted trials are the norm whenever subjects self-select into an intervention, which is often the case across domains stretching from healthcare to advertising. We propose a nonparametric approach to causal inference in non-targeted trials, based on a pre-cluster and merge strategy.

Assume a population is broken into $\ell$ groups with different expected treatment effects in each group. Identify each group with the level of its treatment effect, so there are effect levels $c = 0, 1, \ldots, \ell-1$. For example, a population's subjects can be healthy, $c = 0$, or sick, $c = 1$. We use the Rubin-Neyman potential outcome framework, (19). A subject is a tuple $s = (x, c, t, y)$ sampled from a distribution $D$, where $x \in [0,1]^d$ is a feature-vector such as [age, weight], $c$ indicates the subject's level, $t$ indicates the subjects treatment cohort, and $y$ is the observed outcome. The observed outcome is one of two potential outcomes, $v$ if treated or $\bar{v}$ if not treated. We consider strongly ignorable trials: given $x$, the propensity to treat is strictly between 0 and 1 and the potential outcomes $\{v, \bar{v}\}$ depend only on $x$, independent of $t$. In a strongly ignorable trial, one can use the features to identify counterfactual controls for estimating effect. The level $c$ is central to the scope of our work. Mathematically, $c$ is a hidden effect modifier which determines the distribution of the potential outcomes ($c$ is an unknown and possibly complex function of $x$). The level $c$ dichotomizes the feature space into subpopulations with different effects. One tries to design the eligibility criteria for the trial to ensure that the propensity to treat is non-zero only for subjects in one level. What to do when the eligibility criteria allow more than one level into the trial is exactly the problem we address. Though our work applies to a general number of levels, all the main ideas can be illustrated with just two levels, $c \in \{0, 1\}$. For the sake of concreteness, we denote these two levels healthy and sick.

A trial samples $n$ subjects, $s_1, \ldots, s_n$. If subject $i$ is treated, $t_i = 1$ and the observed outcome $y_i = v_i$, otherwise $t_i = 0$, and the observed outcome is $\bar{v}_i$ (consistency). The treated group is $\mathcal{T} = \{i \mid t_i = 1\}$, the control group is $\mathcal{C} = \{i \mid t_i = 0\}$, and the sick group is $\mathcal{S} = \{i \mid c_i = 1\}$. Our task is to determine if the treatment works on the sick, and if there is any side-effect on the healthy. We wish to estimate the effect and side-effect, defined as

$$\begin{aligned} \text{EFF} &= \mathbb{E}_D[v - \bar{v} \mid c = 1] \\ \text{SIDE-EFF} &= \mathbb{E}_D[v - \bar{v} \mid c = 0]. \end{aligned} \tag{1}$$

Most prior work estimates EFF using the average treatment effect for the treated, the ATT (1),

$$\text{ATT} = \text{average}_{i \in \mathcal{T}}(v_i) - \text{average}_{i \in \mathcal{T}}(\bar{v}_i), \tag{2}$$

which assumes all treated subjects are sick. There are several complications with this approach.

(i) Suppose a subject is treated with probability $p(x, c)$, the propensity to treat. For a non-uniform propensity to treat, the treated group has a selection bias, and ATT is a biased estimate of EFF. Ways to address this bias include inverse propensity weighting, (18), matched controls, (1), and learning the outcome function $y(x, t)$, see for example (2; 3; 10; 12; 22; 23). Alternatively, one can simply ignore this bias and accept that ATT is estimating $\mathbb{E}[v - \bar{v} \mid t = 1]$.

(ii) The second term on the RHS in (2) can't be computed because we don't know the counterfactual $\bar{v}$ for treated subjects. Much of causal inference deals with accurate unbiased estimation of $\text{average}_{i \in \mathcal{T}}(\bar{v}_i)$, (4; 9). Our goal is not to improve counterfactual estimation. Hence, in our experiments, we use off-the-shelf counterfactual estimators.

(iii) (*Focus of our work*) The trial is non-targeted and some (often most) treated subjects are healthy.

To highlight the challenge in (iii) above, consider a simple case with uniform propensity to treat, $p(x, c) = p$. Conditioning on at least one treated subject,

$$\mathbb{E}[\text{ATT}] = \mathbb{P}[\text{sick}] \times \text{EFF} + \mathbb{P}[\text{healthy}] \times \text{SIDE-EFF}.$$

The ATT is a mix of effect and side effect and is therefore biased when the treatment effect is heterogeneous across levels. In many settings, for example healthcare, $\mathbb{P}[\text{sick}] \ll \mathbb{P}[\text{healthy}]$ and the bias is extreme, rendering ATT useless. Increasing the number of subjects won't resolve this bias. State-of-the-art causal inference packages provide methods to compute ATT, specifically aimed at accurate estimates of the counterfactual $\text{average}_{i \in \mathcal{T}}(\bar{v}_i)$, (5; 21). These packages suffer from the mixing bias above. We propose a fix which can be used as an add-on to these packages.

**Our Contribution.** Our main result is an asymptotically consistent distribution independent algorithm to extract the correct effect levels and associated subpopulations in non-targeted trials, when the number of effect-levels is *unknown*. Our main result is Theorem 1. Assume a non-targeted trial has a treated group with $n$ subjects sampled from an unknown distribution $D$. There is an algorithm which identifies $\hat{\ell}$ effect-levels with estimated expected effect $\hat{\mu}_c$ in level $c$, and assigns each subject $s_i$ to a level $\hat{c}_i$ which, under mild technical conditions, satisfies:

**Theorem 1.** *All of the following hold with probability $1 - o(1)$:*

*(1) $\hat{\ell} = \ell$, i.e., the correct number of effect levels $\ell$ is identified.*
*(2) $\hat{\mu}_c = \mathbb{E}[v - \bar{v} \mid c] + o(1)$, i.e., the effect at each level is estimated accurately.*
*(3) The fraction of subjects assigned the correct effect level is $1 - o(1)$. The effect level $\hat{c}_i$ is correct if $\mu_{\hat{c}_i}$ matches, to within $o(1)$, the expected treatment effect for the subject.*

For the formal assumptions, see Section 3. Parts (1) and (2) say the algorithm extracts the correct number of levels and their expected effects. Part (3) says the correct subpopulations for each level are extracted. Knowing the correct subpopulations is useful for post processing, for example to understand the effects in terms of the features. Our algorithm satisfying Theorem 1 is given in Section 2. The algorithm uses an unsupervised pre-cluster and merge strategy which reduces the task of estimating the effect-levels to a 1-dimensional optimal clustering problem that provably extracts the correct levels asymptotically as $n \to \infty$. Our algorithm assumes an unbiased estimator of counterfactuals, for example some established method (5; 21). In practice, this means one can control for confounders. If unbiased counterfactual estimation is not possible, then any form of causal effect analysis is doomed. Our primary goal is untangling the heterogeneous effect levels, hence we use an off-the-shelf gradient boosting algorithm to get counterfactuals in our experiments (5).

We demonstrate that our algorithm's performance on synthetic data matches the theory. Subpopulation effect-analysis is a special case of heterogeneous treatment effects (HTE), (12; 20; 23). Hence, we also compare with X-Learner, a state-of-the art algorithm for HTE (12) and Bayes optimal prediction of effect-level. In comparison to X-Learner, our algorithm extracts visually better subpopulations, and has an accuracy that is more than $10\times$ better for estimating per-subject expected effects. Note, HTE algorithms do not extract subpopulations with effect-levels. They predict effect given the features $x$. One can, however, try to infer subpopulations from predicted effects. Our algorithm also significantly outperforms Bayes optimal based on individual effects, which suggests that some form of pre-cluster and merge strategy is necessary. This need for some form of clustering has been independently observed in (11, chapter 4) who studies a variety of clustering approaches in a non-distribution independent setting with a known number of levels.

## 2  ALGORITHM: PRE-CLUSTER AND MERGE FOR SUBPOPULATION EFFECTS (PCM)

Our algorithm uses a nonparametric pre-cluster and merge strategy that achieves asymptotic consistency without any user-specified hyperparameters. The inputs are the $n$ subjects $s_1, \ldots, s_n$, where

$$\{s_i\}_{i=1}^n = \{(x_i, t_i, y_i, \bar{y}_i)\}_{i=1}^n.$$

Note, both the factual $y_i$ and counterfactual $\bar{y}_i$ are inputs to the algorithm. To use the algorithm in practice, of course, the counterfactual must be estimated, and for our demonstrations we use an out-of-the-box gradient boosting regression algorithm from (7; 16) to estimate counterfactuals. Inaccuracy in counterfactual estimation will be accommodated in our analysis. The need to estimate counterfactuals does impact the algorithm in practice, due to an asymmetry in most trials: the treated population is much smaller than the controls. Hence, one might be able to estimate counterfactuals for the treated population but not for the controls due to lack of coverage by the (small) treated population. In this case, our algorithm is only run on the treated population. It is convenient to define individual treatment effects $\text{ITE}_i = (y_i - \bar{y}_i)(2t_i - 1)$, where $y_i$ is the observed factual and $\bar{y}_i$ the counterfactual ($2t_i - 1 = \pm 1$ ensuring that the effect computed is for treatment versus no treatment). There are five main steps.

---

1: [PRE-CLUSTER] Cluster the $x_i$ into $K \in O(\sqrt{n})$ clusters $Z_1, \ldots, Z_K$.

2: Compute ATT for each cluster $Z_j$, $\text{ATT}_j = \text{average}_{x_i \in Z_j} \text{ITE}_i$.

3: [MERGE] Group the $\{\text{ATT}_j\}_{j=1}^K$ into $\hat{\ell}$ effect-levels, merging the clusters at each level to get subpopulations $X_0, X_1, \ldots, X_{\hat{\ell}-1}$. ($X_c$ is the union of all clusters at level $c$.)

4: Compute subpopulation effects $\hat{\mu}_c = \text{average}_{x_i \in X_c} \text{ITE}_i$, for $c = 0, \ldots, \hat{\ell} - 1$.

5: Assign subjects to effect levels, update the populations $X_c$ and expected effects $\hat{\mu}_c$.

---

We now elaborate on the intuition and details for each step in the algorithm.

**Step 1.** The clusters in the pre-clustering step play two roles. The first is to denoise individual effects using in-cluster averaging. The second is to group like with like, that is clusters should be homogeneous, containing only subjects from one effect-level. This means each cluster-ATT will accurately estimate a single level's effect (we do not know which). We allow for any clustering algorithm. However, our theoretical analysis (for simplicity) uses a specific algorithm, box-clustering, based on an $\varepsilon$-net of the feature space. One could also use a standard clustering algorithm such as $K$-means. We compare box-clustering with $K$-means in the appendix.

**Step 2.** Denoising of the individual effects using in-cluster averaging. Assuming clusters are homogeneous, each cluster ATT will approximate some level's effect.

**Step 3.** Assuming the effects in different levels are well separated, this separation gets emphasized in the cluster-ATTs, provided clusters are homogeneous. Hence, we can identify effect-levels from the clusters with similar effects, and merge those clusters into subpopulations. Two tasks must be solved. Finding the number of subpopulations $\hat{\ell}$ and then optimally grouping the clusters into $\hat{\ell}$ subpopulations. To find the subpopulations, we use $\hat{\ell}$-means with squared 1-dim clustering error.

Our algorithm sets $\hat{\ell}$ to achieve an $\hat{\ell}$-means error at most $\log n/n^{1/2d}$. So,

$$\text{optimal 1-dim clustering error}(\hat{\ell} - 1) > \log n/n^{1/2d}$$
$$\text{optimal 1-dim clustering error}(\hat{\ell}) \leq \log n/n^{1/2d}$$

Simultaneously finding $\hat{\ell}$ and optimally partitioning the clusters into $\hat{\ell}$ groups can be solved using a standard dynamic programming algorithm in $O(K^2\hat{\ell})$ time using $O(K)$ space (24).

Note, our algorithm will identify the number of effect levels provided such distinct subpopulations exist in the data. If it is known that only two subpopulations exist, sick and healthy, then $\hat{\ell}$ can be hard-coded to 2.

**Step 4.** Assuming each cluster is homogeneous and clusters with similar effects found in step 3 are from the same effect-level, the subpopulations formed by merging the clusters with similar effects will be nearly homogeneous. Hence, the subpopulation-ATTs will be accurate estimates of the effects at each level.

**Step 5.** Each subject $x_i$ is implicitly assigned a level $\hat{c}_i$ based on the subpopulation $X_c$ to which it belongs. However, we can do better. By considering the $\sqrt{n}$ nearest neighbors to $x_i$, we can obtain a smoothed effect for $x_i$. We use this smoothed effect to place $x_i$ into the subpopulation whose effect matches best, hence placing $x_i$ into a level. Unfortunately, running this algorithm for all $n$ subjects is costly, needing sophisticated data structures to reduce the expected run time below $O(n^2)$. As an alternative, we center an $(1/n^{1/2d})$-hypercube on $x_i$ and smooth $x_i$'s effect using the average effect over points in this hypercube. This approach requires $O(n\sqrt{n})$ run time to obtain the effect-level for all subjects, significantly better than $O(n^2)$ when $n$ is large. Once the effect-levels for all subjects are obtained, one can update the subpopulations $X_c$ and the corresponding effect-estimates $\hat{\mu}_c$.

The run time of the algorithm is $O(n\ell + n\sqrt{n})$ (expected and with high probability) and the output is nearly homogeneous subpopulations which can now be post-processed. An example of useful post-processing is a feature-based explanation of the subpopulation-memberships. Note that we still do not know which subpopulation(s) are the sick ones, hence we cannot say which is the effect and which is the side effect. A post-processing oracle would make this determination. For example, a doctor in a medical trial would identify the sick groups from subpopulation-demographics.

**Note.** The optimal 1-d clustering can be done directly on the smoothed ITEs from the $(1/n^{1/2d})$-hypercubes centered on each $x_i$, using the same thresholds in step 3. One still gets asymptotic consistency, however the price is an increased run time to $O(n^2\ell)$. This is prohibitive for large $n$.

## 3    ASYMPTOTIC CONSISTENCY: PROOF OF THEOREM 1

To prove consistency, we must make our assumptions precise. In some cases the assumptions are stronger than needed, for simplicity of exposition.

**A1.** The feature space $X$ is $[0,1]^d$ and the marginal feature-distribution is uniform, $D(x) = 1$. More generally, $X$ is compact and $D(x)$ is bounded, $0 < \delta \leq D(x) \leq \Delta$ (can be relaxed).

**A2.** The level $c$ is an unknown function of the feature $x$, $c = h(x)$. Potential effects depend only on $c$. Conditioning on $c$, effects are well separated. Let $\mu_c = \mathbb{E}_D[v - \bar{v}|c]$. Then,

$$|\mu_c - \mu_{c'}| \geq \kappa \qquad \text{for } c \neq c'$$

**A3.** Define the subpopulation for level $c$ as $X_c = h^{-1}(c)$. Each subpopulation has positive measure, $\mathbb{P}[x \in X_c] = \beta_c \geq \beta > 0$.

**A4.** For a treated subject $x_i$ with outcome $y_i$, it is possible to produce an unbiased estimate of the counterfactual outcome $\bar{y}_i$. Effectively, we are assuming an unbiased estimate of the individual treatment effect $\text{ITE}_i = y_i - \bar{y}_i$ is available. Any causality analysis requires some estimate of counterfactuals and, in practice, one typically gets counterfactuals from the untreated subjects after controlling for confounders (5; 21).

**A5.** Sample averages concentrate. Essentially, the estimated ITEs are independent. This is true in practice because the subjects are independent and the counterfactual estimates use a predictor learned from the independent control population. For $m$ i.i.d. subjects, let the average of the estimated ITEs be $\hat{\nu}$ and the expectation of this average be $\nu$. Then,

$$\mathbb{P}[|\hat{\nu} - \nu| > \epsilon] \leq e^{-\gamma m \epsilon^2}.$$

The parameter $\gamma > 0$ is related to distributional properties of the estimated ITEs. Higher variance ITE estimates result in $\gamma$ being smaller. Concentration is a mild technical assumption requiring the estimated effects to be unbiased well behaved random variables, to which a central limit theorem applies. Bounded effects or normally distributed effects suffice for concentration.

**A6.** The boundary between the subpopulations has small measure. Essentially we require that two subjects that have very similar features will belong to the same level with high probability (the function $c = h(x)$ is not a "random" function). Again, this is a mild technical assumption which is taken for granted in practice. Let us make the assumption more precise. Define an $\varepsilon$-net to be a subdivision of $X$ into $(1/\varepsilon)^d$ disjoint hypercubes of side $\varepsilon$. A hypercube of an $\varepsilon$-net is impure if it contains points from multiple subpopulations. Let $N_{\text{impure}}$ be the number of impure hypercubes in an $\varepsilon$-net. Then $\varepsilon^d N_{\text{impure}} \leq \alpha \varepsilon^\rho$, where $\rho > 0$ and $\alpha$ is a constant. Note, $d - \rho$ is the boxing-dimension of the boundary. In most problems, $\rho = 1$.

**A7.** We use box-clustering for the first step in the algorithm. Given $n$, define $\varepsilon(n) = 1/\lfloor n^{1/2d} \rfloor$. All points in a hypercube of an $\varepsilon(n)$-net form a cluster. Note that the number of clusters is approximately $\sqrt{n}$. The expected number of points in a cluster is $n\varepsilon(n)^d \approx \sqrt{n}$.

We prove Theorem 1 via a sequence of lemmas. The feature space $X = [0, 1]^d$ is partitioned into levels $X_0, \ldots, X_{\ell-1}$, where $X_c = h^{-1}(c)$ is the set of points whose level is $c$. Define an $\varepsilon$-net that partitions $X$ into $N_\varepsilon = \varepsilon^{-d}$ hypercubes of equal volume $\varepsilon^d$, where $\varepsilon$ is the side-length of the hypercube. Set $\varepsilon = 1/\lfloor n^{1/2d} \rfloor$. Then, $N_\varepsilon = \sqrt{n}(1 - O(d/n^{1/2d})) \sim \sqrt{n}$. Each hypercube in the $\varepsilon$-net defines a cluster for the pre-clustering stage. There are about $\sqrt{n}$ clusters and, since $D(x)$ is uniform, there are about $\sqrt{n}$ points in each cluster. Index the clusters in the $\varepsilon$-net by $j \in \{1, \ldots, N_\varepsilon\}$ and define $n_j$ as the number of points in cluster $j$. Formally, we have,

**Lemma 1.** *Suppose $D(x) \geq \delta > 0$. Then, $\mathbb{P}[\min_j n_j \geq \frac{1}{2}\delta\sqrt{n}] > 1 - \sqrt{n}\exp(-\delta\sqrt{n}/8)$.*

*Proof.* Fix a hypercube in the $\varepsilon$-net. Its volume is $\varepsilon^d \geq (1/n^{1/2d})^d = 1/\sqrt{n}$. A point lands in this hypercube with probability at least $\delta/\sqrt{n}$. Let $Y$ be the number of points in the hypercube. Then, $Y$ is a sum of $n$ independent Bernoullis and $\mathbb{E}[Y] \geq \delta\sqrt{n}$. By a Chernoff bound (14, page 70),

$$\mathbb{P}[Y < \delta\sqrt{n}/2] \leq \mathbb{P}[Y < \mathbb{E}[Y]/2] < \exp(-\mathbb{E}[Y]/8) \leq \exp(-\delta\sqrt{n}/8).$$

By a union bound over the $N_\varepsilon$ clusters,

$$\mathbb{P}[\text{some cluster has fewer than } \delta\sqrt{n}/2 \text{ points}] < N_\varepsilon \exp(-\delta\sqrt{n}/8) \leq \sqrt{n}\exp(-\delta\sqrt{n}/8).$$

The lemma follows by taking the complement event. ∎

For uniform $D(x)$, $\delta = 1$ and every cluster has at least $\frac{1}{2}\sqrt{n}$ points with high probability. We can now condition on this high probability event that every cluster is large. This means that a cluster's ATT is an average of many ITEs, which by **A5** concentrates at the expected effect for the hypercube. Recall that the expected effect in level $c$ is defined as $\mu_c = \mathbb{E}_D[v - \bar{v}|c]$. We can assume, w.l.o.g., that $\mu_0 < \mu_1 \cdots < \mu_{\ell-1}$. Define $\nu_j$ as the expected average effect for points in the hypercube $j$ and $\text{ATT}_j$ as the average ITE for points in cluster $j$. since every cluster is large, every cluster's $\text{ATT}_j$ will be close to its expected average effect $\nu_j$. More formally,

**Lemma 2.** $\mathbb{P}[\max_j |\text{ATT}_j - \nu_j| \leq 2\sqrt{\log n/\gamma\delta\sqrt{n}}] \geq 1 - n^{-3/2} - \sqrt{n}\exp(-\delta\sqrt{n}/8)$.

*Proof.* Conditioning on $\min_j n_j \geq \frac{1}{2}\delta\sqrt{n}$ and using **A5**, we have

$$\mathbb{P}\left[|\text{ATT}_j - \nu_j| > 2\sqrt{\log n/\gamma\delta\sqrt{n}} \,\Big|\, \min_j n_j \geq \tfrac{1}{2}\delta\sqrt{n}\right] \leq \exp(-2\log n) = 1/n^2.$$

By a union bound, $\mathbb{P}[\max_j |\text{ATT}_j - \nu_j| > 2\sqrt{\log n/\gamma\delta\sqrt{n}} \mid \min_j n_j \geq \frac{1}{2}\delta\sqrt{n}] \leq N_\varepsilon/n^2$. For any events $\mathcal{A}, \mathcal{B}$, by total probability, $\mathbb{P}[\mathcal{A}] \leq \mathbb{P}[\mathcal{A} \mid \mathcal{B}] + \mathbb{P}[\overline{\mathcal{B}}]$. Therefore,

$$\mathbb{P}[\max_j |\text{ATT}_j - \nu_j| > 2\sqrt{\log n/\gamma\delta\sqrt{n}}] \leq N_\varepsilon/n^2 + \mathbb{P}[\min_j n_j < \tfrac{1}{2}\delta\sqrt{n}]$$

To conclude the proof, use $N_\varepsilon \leq \sqrt{n}$ and Lemma 1. ∎

A hypercube in the $\varepsilon$-net is homogeneous if it only contains points of one level (the hypercube does not intersect the boundary between levels). Let $N_c$ be the number of homogeneous hypercubes for level $c$ and $N_{\text{impure}}$ be the number of hypercubes that are not homogeneous, i.e., impure.

**Lemma 3.** $N_{impure} \leq \alpha\varepsilon^\rho N_\varepsilon$ and $N_c \geq N_\varepsilon(\beta/\Delta - \alpha\varepsilon^\rho)$.

*Proof.* **A6** directly implies $N_{\text{impure}} \leq \alpha\varepsilon^\rho N_\varepsilon$. Only the pure level $c$ or impure hypercubes can contain points in level $c$. Using **A3** and $\varepsilon^d = 1/N_\varepsilon$, we have

$$\beta \leq \mathbb{P}[x \in X_c] \leq (N_c + N_{\text{impure}})\Delta\varepsilon^d \leq (N_c + \alpha\varepsilon^\rho N_\varepsilon)\Delta/N_\varepsilon.$$

The result follows after rearranging the above inequality. ∎

The main tools we need are Lemmas 2 and 3. Let us recap what we have. The cluster ATTs are close to the expected average effect in every hypercube. The number of impure hypercubes is an asymptotically negligible fraction of the hypercubes since $\varepsilon \in O(1/n^{1/2d})$. Each level has an asymptotically constant fraction of homogeneous hypercubes. This means that almost all cluster ATTs will be close to a level's expected effect, and every level will be well represented. Hence, if we optimally cluster the ATTs, with fewer than $\ell$ clusters, we won't be able to get clustering error close to zero. With at least $\ell$ clusters, we will be able to get clustering error approaching zero. This is the content of the next lemma, which justifies step 3 in the algorithm. An optimal $k$-clustering of the cluster ATTs produces $k$ centers $\theta_1, \ldots, \theta_k$ and assigns each cluster $\text{ATT}_j$ to a center $\theta(\text{ATT}_j)$ so that the average clustering error $\text{err}(k) = \sum_j (\text{ATT}_j - \theta(\text{ATT}_j))^2/N_\varepsilon$ is minimized. Given $k$, one can find an optimal $k$-clustering in $O(N_\varepsilon^2 k)$ time using $O(N_\varepsilon)$ space.

**Lemma 4.** *With probability at least $1 - n^{-3/2} - \sqrt{n}\exp(-\delta\sqrt{n}/8)$, optimal clustering of the ATTs with $\ell - 1$ and $\ell$ clusters produces clustering errors which satisfy*

$$\begin{aligned}
\text{err}(\ell - 1) &\geq (\beta/\Delta - \alpha\epsilon^\rho)\left(\kappa/2 - 2\sqrt{\log n/\gamma\delta\sqrt{n}}\right)^2 && \text{for } \frac{\log n}{\sqrt{n}} < \frac{\kappa^2\gamma\delta}{16} \\
\text{err}(\ell) &\leq \tfrac{1}{4}\alpha\varepsilon^\rho(\mu_{\ell-1} - \mu_0)^2 + 4\log n(1 + \alpha\varepsilon^\rho)/\gamma\delta\sqrt{n}
\end{aligned}$$

*Proof.* With the stated probability, by Lemma 2, all ATTs are within $2\sqrt{\log n/\gamma\delta\sqrt{n}}$ of the expected effect for their respective hypercube. This, together with Lemma 3 is enough to prove the bounds.

First, the upper bound on $\text{err}(\ell)$. Choose cluster centers $\mu_0, \ldots, \mu_{\ell-1}$, the expected effect for each level. This may not be optimal, so it gives an upper bound on the cluster error. Each homogeneous hypercube has a expected effect which is one of these levels, and its ATT is within $2\sqrt{\log n/\gamma\delta\sqrt{n}}$ of the corresponding $\mu$. Assign each ATT for a homogeneous hypercube to its corresponding $\mu$. The homogeneous hypercubes have total clustering error at most $4\log n(N_\varepsilon - N_{\text{impure}})/\gamma\delta\sqrt{n}$. For an impure hypercube, the expected average effect is a convex combination of $\mu_0, \ldots, \mu_{\ell-1}$. Assign these ATTs to either $\mu_0$ or $\mu_{\ell-1}$, with an error at most $(2\sqrt{\log n/\gamma\delta\sqrt{n}} + \tfrac{1}{2}(\mu_{\ell-1} - \mu_0))^2$. Thus,

$$\begin{aligned}
N_\varepsilon\text{err}(\ell) &\leq \frac{4\log n(N_\varepsilon - N_{\text{impure}})}{\gamma\delta\sqrt{n}} + N_{\text{impure}}(2\sqrt{\log n/\gamma\delta\sqrt{n}} + \tfrac{1}{2}(\mu_{\ell-1} - \mu_0))^2 \\
&\leq \frac{4\log n(N_\varepsilon + N_{\text{impure}})}{\gamma\delta\sqrt{n}} + \frac{N_{\text{impure}}(\mu_{\ell-1} - \mu_0)^2}{2}
\end{aligned}$$

The upper bound follows after dividing by $N_\varepsilon$ and using $N_{\text{impure}} \leq \alpha\varepsilon^\rho N_\varepsilon$.

Now, the lower bound on $\text{err}(\ell - 1)$. Consider any $\ell - 1$ clustering of the ATTs with centers $\theta_0, \ldots, \theta_{\ell-2}$. At least $N_c \geq N_\varepsilon(\beta/\Delta - \alpha\epsilon^\rho)$ of the ATTs are within $2\sqrt{\log n/\gamma\delta\sqrt{n}}$ of $\mu_c$. We also know that $\mu_{c+1} - \mu_c \geq \kappa$. Consider the $\ell$ disjoint intervals $[\mu_c - \kappa/2, \mu_c + \kappa/2]$. By the pigeonhole principle, at least one of these intervals $[\mu_{c*} - \kappa/2, \mu_{c*} + \kappa/2]$ does not contain a center. Therefore all the ATTs associated to $\mu_{c*}$ will incur an error at least $\kappa/2 - 2\sqrt{\log n/\gamma\delta\sqrt{n}}$ when $\kappa/2 > 2\sqrt{\log n/\gamma\delta\sqrt{n}}$. The total error is

$$N_\varepsilon\text{err}(\ell - 1) \geq N_{c*}\left(\kappa/2 - 2\sqrt{\log n/\gamma\delta\sqrt{n}}\right)^2.$$

Using $N_{c*} \geq N_\varepsilon(\beta/\Delta - \alpha\epsilon^\rho)$ and dividing by $N_\varepsilon$ concludes the proof. ∎

Lemma 4 is crucial to estimating the number of levels. The error is $\beta\kappa^2/4\Delta(1+o(1))$ for fewer than $\ell$ clusters and $\frac{1}{4}\alpha\varepsilon^\rho(\mu_{\ell-1}-\mu_0)^2(1+o(1))$ for $\ell$ or more clusters. Any function $\tau(n)$ that asymptotically separates these two errors can serve as an error threshold. The function should be agnostic to the parameters $\alpha,\beta,\kappa,\Delta,\rho,\ldots$. In practice, $\rho=1$ and since $\varepsilon\sim 1/n^{1/2d}$, we have chosen $\tau(n)=\log n/n^{\rho/2d}$. Since $\text{err}(\ell-1)$ is asymptotically constant, $\ell-1$ clusters can't achieve error $\tau(n)$ (asymptotically). Since $\text{err}(\ell)\in O(\varepsilon^\rho)$, $\ell$ clusters can achieve error $\tau(n)$ (asymptotically). Hence, choosing $\hat{\ell}$ as the minimum number of clusters that achieves error $\tau(n)$ will asymptotically output the correct number of clusters $\ell$, with high probability, proving part (1) of Theorem 1.

We now prove parts (2) and (3) of Theorem 1, which follow from the accuracy of steps 4 and 5 in the algorithm. We know the algorithm asymptotically selects the correct number of levels with high probability. We show that each level is populated by mostly the homogeneous clusters of that level.

**Lemma 5.** *With probability at least $1-n^{-3/2}-\sqrt{n}\exp(-\delta\sqrt{n}/8)$, asymptotically in $n$, all the $N_c$* ATT*s from the homogeneous hypercubes of level $c$ are assigned to the same cluster in the optimal clustering, and no* ATT*s from a different level's homogeneous hypercubes is assigned to this cluster.*

*Proof.* Similar to the proof of Lemma 4, consider the $\ell$ disjoint intervals $[\mu_c-\kappa/4,\mu_c+\kappa/4]$. One center $\theta_c$ must be placed in this interval otherwise the clustering error is asymptotically constant, which is not optimal. All the ATTs for level $c$ are (as $n$ gets large) more than $\kappa/2$ away from any other center, and at most $\kappa/2$ away from $\theta_c$, which means all these ATTs get assigned to $\theta_c$. ∎

Similar to Lemma 1, we can get a high-probability upper bound of $a\sqrt{n}$ on the maximum number of points in a cluster. Asymptotically, the number of points in the impure clusters is $n_{\text{impure}}\in O(\varepsilon^\rho\sqrt{n}N_\varepsilon)$. Suppose these impure points have expected average effect $\mu$ (a convex combination of the $\mu_c$'s). The number of points in level $c$ homogeneous clusters is $n_c\in\Omega(\sqrt{n}N_\varepsilon)$. Even if all impure points are added to level $c$, the expected average effect for the points in level $c$ is

$$\mathbb{E}[\text{ITE}\mid\text{assigned to level } c]=\frac{n_{\text{impure}}\mu+n_c\mu_c}{n_{\text{impure}}+n_c}=\mu_c+O(\varepsilon^\rho). \qquad (3)$$

Part (2) of Theorem 1 follows from the next lemma after setting $\varepsilon\sim 1/n^{1/2d}$ and $\rho=1$.

**Lemma 6.** *Estimate $\hat{\mu}_c$ as the average* ITE *for all points assigned to level $c$ (the $c$th order statistic of the optimal centers $\theta_0,\ldots,\theta_{\hat{\ell}-1}$). Then $\hat{\mu}_c=\mu_c+O(\varepsilon^\rho+\sqrt{\log n/n})$ with probability $1-o(1)$.*

*Proof.* Apply a Chernoff bound. We are taking an average of proportional to $n$ points with expectation in (3). This average will approximate the expectation to within $\sqrt{\log n/n}$ with probability $1-o(1)$. The details are very similar to the proof of Lemma 2, so we omit them. ∎

Part (3) of Theorem 1 now follows because all but the $O(\varepsilon^\rho)$ fraction of points in the impure clusters are assigned a correct expected effect. An additional fine-tuning leads to as much as $2\times$ improvement in experiments. For each point, consider the $\varepsilon$-hypercube centered on that point. By a Chernoff bound, each of these $n$ hypercubes has $\Theta(\sqrt{n})$ points, as in Lemma 1. All but a fraction $O(\varepsilon^\rho)$ of these are impure. Assign each point to the center $\theta_c$ that best matches its hypercube-"smoothed" ITE, giving new subpopulations $X_c$ and corresponding subpopulation-effects $\hat{\mu}_c$. This EM-style update can be iterated. Our simulations show the results for one E-M update.

## 4 DEMONSTRATION ON SYNTHETIC DATA

We use a 2-dimensional synthetic experiment with three levels to demonstrate our pre-cluster and merge algorithm (PCM). Alternatives to pre-clustering include state-of-the-art methods that directly predict the effect such as meta-learners, and the Bayes optimal classifier based on ITEs. All methods used a base gradient boosting forest with 400 trees to estimate counterfactuals. The subpopulations in our experiment are shown in Figure 1, where black is effect-level 0, gray is level 1 and white is level 2. We present detailed results with $n=200K$. Extensive results can be found in the appendix. Let us briefly describe the two existing benchmarks we will compare against.

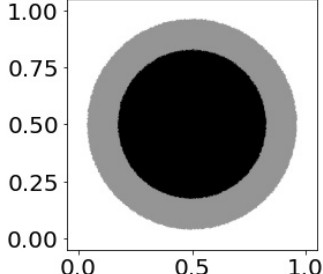

The treatment $t$ is distributed randomly between the subjects. The outcome $y$, conditioned on $c$ and $t$, is Gaussian with std. dev. 5:

$$y(t,c) \sim N(\mu_{(t,c)}, 5)$$

The three sub-populations have treatment effects of 0,1,2. The expected potential outcome for treatment and level $(t,c)$ are:

$$\mu_{(0,0)} = 0 \qquad \mu_{(1,0)} = 0,$$
$$\mu_{(0,1)} = 0 \qquad \mu_{(1,1)} = 1,$$
$$\mu_{(0,2)} = 0 \qquad \mu_{(1,2)} = 2.$$

Figure 1: Subpopulations for synthetic data.

**X-learner (12)**, is a meta-learner that estimates heterogeneous treatment effects directly from ITEs. For the outcome and effect models of X-Learner we use a base gradient boosting learner with 400 estimators (6) implemented in scikit-learn (16). For the propensity model we use logistic regression.

**Bayes Optimal** uses the ITEs to reconstruct the subpopulations, given the number of levels and the ground-truth outcome distribution $y(t,c)$ from Figure 1. The Bayes optimal classifier is: $c_{\text{Bayes}} = 0$ if ITE $\leq 0.5$, $c_{\text{Bayes}} = 1$ if $0.5 < $ ITE $\leq 1.5$, $c_{\text{Bayes}} = 2$ if $1.5 < $ ITE. We also use these thresholds to reconstruct subpopulations for X-learner's predicted ITEs. **Note: Neither the thresholds nor the number of levels are available in practice.** We compare the benchmark subpopulations reconstructed with these thresholds to further showcase the power of our algorithm's subpopulations, which outperform the competition without access to the forbidden information.

Let $c_i$ be the level of subject $i$ and $\widehat{\text{ITE}}_i$ the estimated ITE. The error is $|\mu_{c_i} - \widehat{\text{ITE}}_i|$, and we report the mean absolute error in the table below. Our algorithm predicts a level $\hat{c}_i$ and uses its associated effect $\hat{\mu}_{\hat{c}_i}$ as $\widehat{\text{ITE}}_i$. The other methods predict ITE directly for which we compute mean absolute error. As mentioned above, we also show the error for the optimally reconstructed subpopulations, which is not possible in practice, but included for comparison (red emphasizes not available in practice).

| $n$ | PCM (this work) | X-Learner | | Bayes Optimal | |
|---|---|---|---|---|---|
| | | Subpopulations | Predicted-ITE | Subpopulations | Raw-ITE |
| 20K | **0.35±0.39** | $3.04 \pm 1.11$ | $3.07 \pm 2.41$ | $4.57 \pm 1.33$ | $4.59 \pm 3.49$ |
| 200k | **0.109±0.22** | $1.44 \pm 0.83$ | $1.50 \pm 1.38$ | $4.22 \pm 1.28$ | $4.24 \pm 3.22$ |
| 2M | **0.036±0.13** | $0.34 \pm 0.47$ | $0.46 \pm 0.56$ | $4.01 \pm 1.25$ | $4.03 \pm 3.05$ |

Our algorithm is about $10\times$ better than existing benchmarks even though we do not use the forbidden information (number of levels and optimal thresholds). It is also clear that X-learner is significantly better than Bayes optimal with just the raw ITEs. The next table shows subpopulation effects, again red indicates the use of forbidden information on the number of levels and optimal thresholds. The ground truth effects are $\mu_0 = 0, \mu_1 = 1, \mu_2 = 2$.

| $n$ | PCM (this work) | | | X-Learner | | | Bayes Optimal | | |
|---|---|---|---|---|---|---|---|---|---|
| | $\hat{\mu}_0$ | $\hat{\mu}_1$ | $\hat{\mu}_2$ | $\hat{\mu}_0$ | $\hat{\mu}_1$ | $\hat{\mu}_2$ | $\hat{\mu}_0$ | $\hat{\mu}_1$ | $\hat{\mu}_2$ |
| *20K* | -0.21 | 0.91 | 2.07 | -2.5 | 0.99 | 4.44 | -3.94 | 1.00 | 5.99 |
| *200K* | 0.06 | 0.963 | 1.95 | -1.16 | 1.01 | 2.87 | -3.62 | 1.00 | 5.61 |
| *2M* | 0.04 | 0.996 | 1.993 | -0.26 | 0.99 | 2.07 | -3.41 | 1.00 | 5.41 |

Note that $\hat{\mu}_1$ for X-learner and Bayes optimal are accurate, an artefact of knowing the optimal thresholds (not realizable in practice). A detailed comparison of our algorithm (PCM) with X-Learner and Bayes optimal subpopulations is shown in Figure 2. PCM clearly extracts the correct subpopulations. X-Learner and Bayes optimal, even given the number of levels and optimal thresholds, does not come visually close to PCM. Note, X-learner does display some structure but Bayes optimal on just the ITEs is a disaster. This is further illustrated in the ITE-histograms in the second row. PCM clearly shows three levels, where as X-learner ITEs and the raw ITEs suggest just one high variance level. The 3rd row shows the confusion matrices for subpopulation assignment. The red indicates use of information forbidden in practice, however we include it for comparison. The confusion

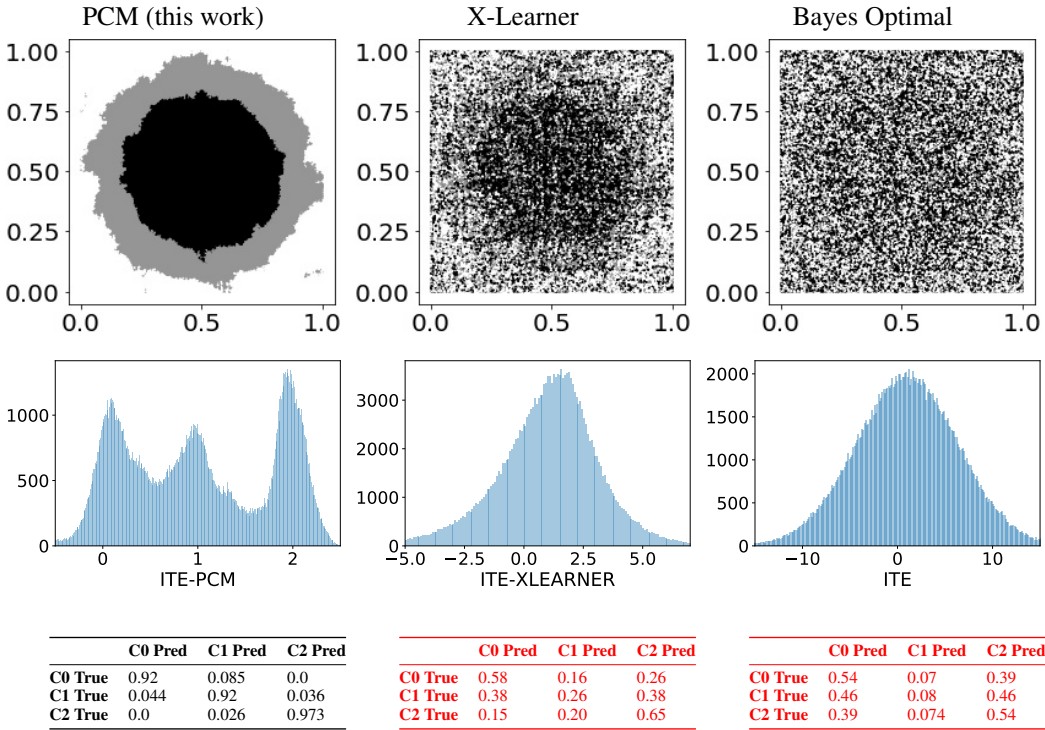

| | C0 Pred | C1 Pred | C2 Pred |
|---|---|---|---|
| **C0 True** | 0.92 | 0.085 | 0.0 |
| **C1 True** | 0.044 | 0.92 | 0.036 |
| **C2 True** | 0.0 | 0.026 | 0.973 |

| | C0 Pred | C1 Pred | C2 Pred |
|---|---|---|---|
| **C0 True** | 0.58 | 0.16 | 0.26 |
| **C1 True** | 0.38 | 0.26 | 0.38 |
| **C2 True** | 0.15 | 0.20 | 0.65 |

| | C0 Pred | C1 Pred | C2 Pred |
|---|---|---|---|
| **C0 True** | 0.54 | 0.07 | 0.39 |
| **C1 True** | 0.46 | 0.08 | 0.46 |
| **C2 True** | 0.39 | 0.074 | 0.54 |

Figure 2: **Top row.** PCM reconstructs superior subpopulations without access to the forbidden information used by X-learner and Bayes optimal (number of levels and optimal thresholds). **Middle row.** The ITE-histogram for PCM clearly shows 3 distinct effects, while the other methods suggest a single high-variance effect. **Bottom Row.** Subpopulation confusion matrices show that PCM extracts the correct subpopulations. The other methods fail even with the forbidden information.

matrix for PCM without forbidden information clearly dominates the other methods which use forbidden information. The high noise in the outcomes undermines the other methods, while PCM is robust. In high noise settings, direct use of the ITEs without some form of pre-clustering fails.

**Summary of experiments with synthetic data.** Our algorithm accurately extracts subpopulations at different effect-levels. Analysis of individual treatment effects fails when there is noise. Our experiments show that practice follows the theory (more detailed experiments, including how cluster homogeneity converges to 1, are shown in the appendix). We note that there is a curse of dimensionality, namely the convergence is at a rate $O(n^{-1/2d})$.

## 5 CONCLUSION

Our work amplifies the realm of causal analysis to non-targeted trials where the treated population can consist of large subpopulations with different effects. Our algorithm uses a plug-and-play pre-cluster and merge strategy that provably untangles the different effects. Experiments on synthetic data show a $10\times$ or more improvement over existing HTE-benchmarks. In our analysis, we did not attempt to optimize the rate of convergence. Optimizing this rate could lead to improved algorithms. Our work allows causal effects analysis to be used in settings such as health interventions, where wide deployment over a mostly healthy population would mask the effect on the sick population. Our methods can seemlessly untangle the effects without knowledge of what sick and healthy mean. This line of algorithms can also help in identifying inequities between the subpopulations. One significant contribution is to reduce the untangling of subpopulation effects to a 1-dim clustering problem which we solve efficently. This approach may be of independent interest beyond causal-effect analysis. The effect is just a function that takes on $\ell$ levels. Our approach can be used to learn any function that takes on a finite number of levels. It could also be used to learn a piecewise approximation to an arbitrary continuous function on a compact set.

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

## A  APPENDIX

We provide more detailed experimental results, specifically results for different $n$ (20K, 200K and 2M) and a comparison of different clustering methods in the pre-clustering phase: box-only, PCM (box plus 1 step of E-M improvement) and K-means. To calculate the counterfactual for treated subjects, we train a gradient boosted forest on the control population.

## B  CONVERGENCE WITH $n$

### B.1  RECONSTRUCTED SUBPOPULATIONS

We show subpopulation reconstructions for $n \in \{20K, 200K, 2M\}$.

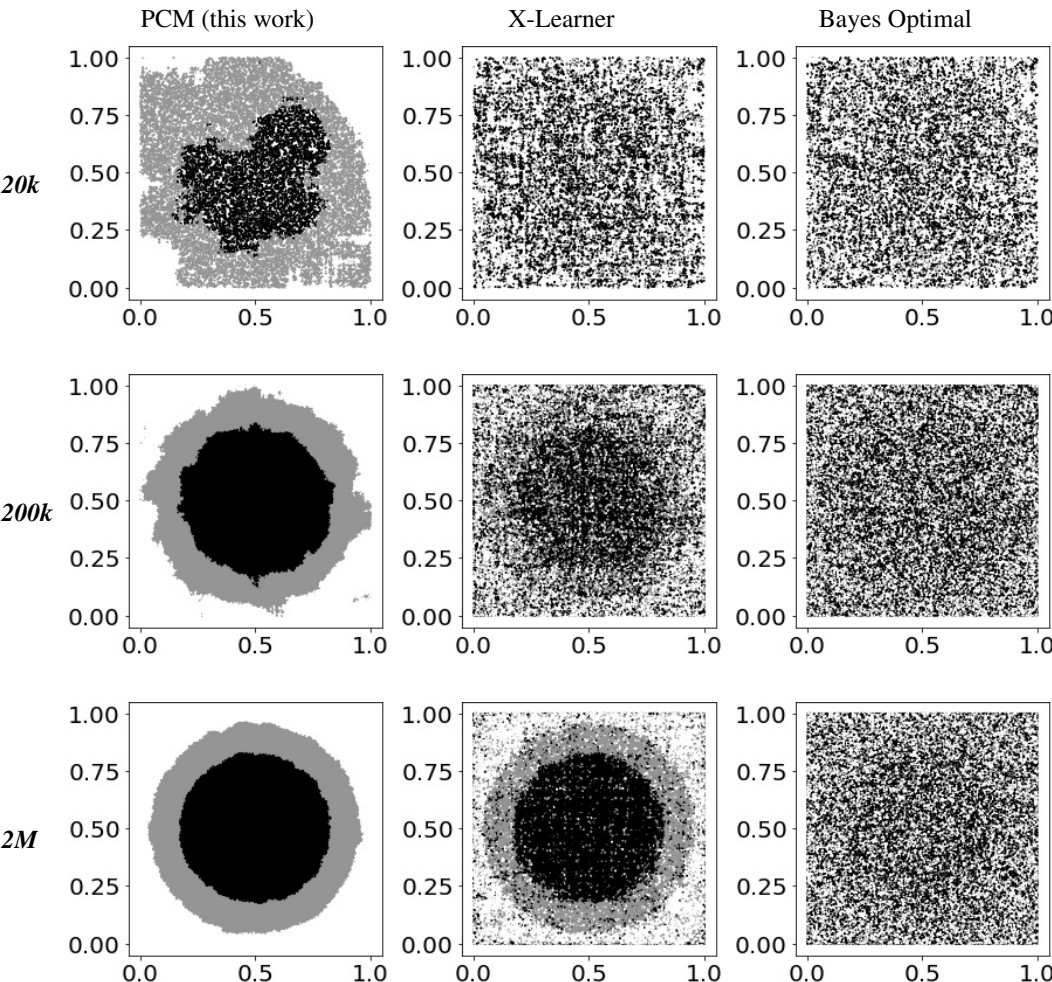

Even with just 20K points in this very noisy setting, PCM is able to extract some meaningful subpopulation structure, while none of the other methods can.

## B.2 ITE HISTOGRAMS

We show the ITE histograms for $n \in \{20K, 200K, 2M\}$.

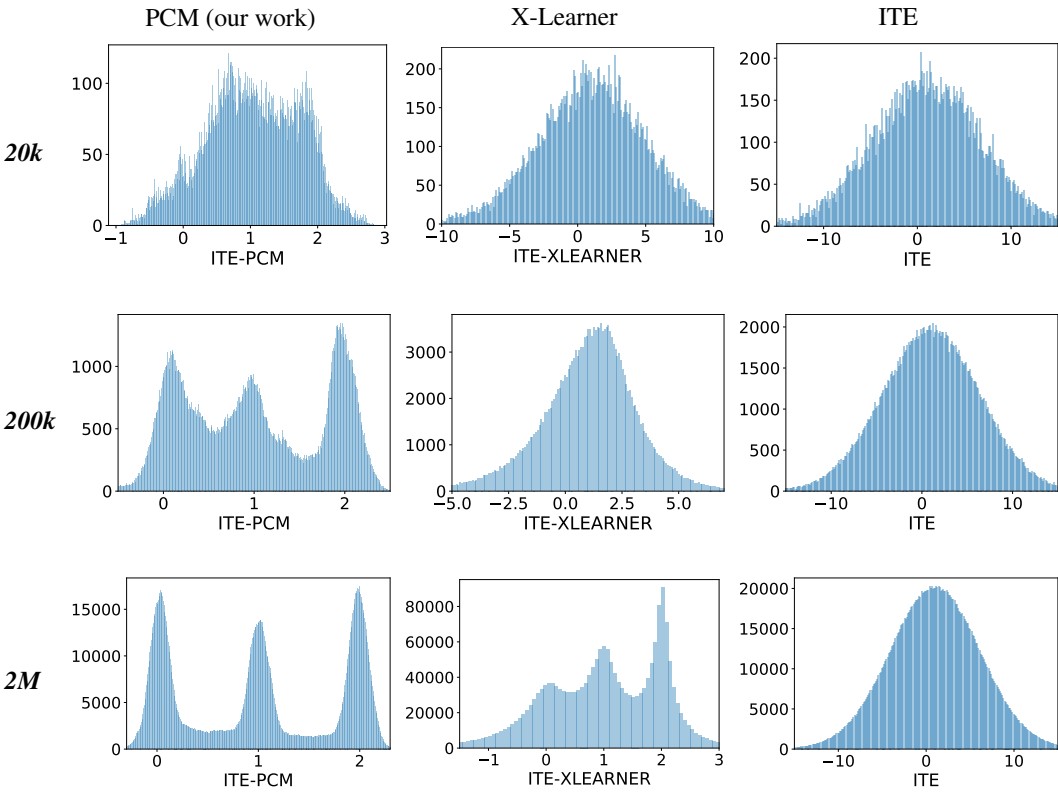

## C   DIFFERENT PRE-CLUSTERING METHODS

We show the reconstructed subpopulations and effect errors for different pre-clustering methods. Box-clustering without any E-M step is also provably consistent. Our algorithm PCM uses box-clustering followed by an E-M step to improve the subpopulations using smoothed ITEs. We also show K-means pre-clustering, for which we did not prove any theoretical guarantees.

**Reconstruction.**

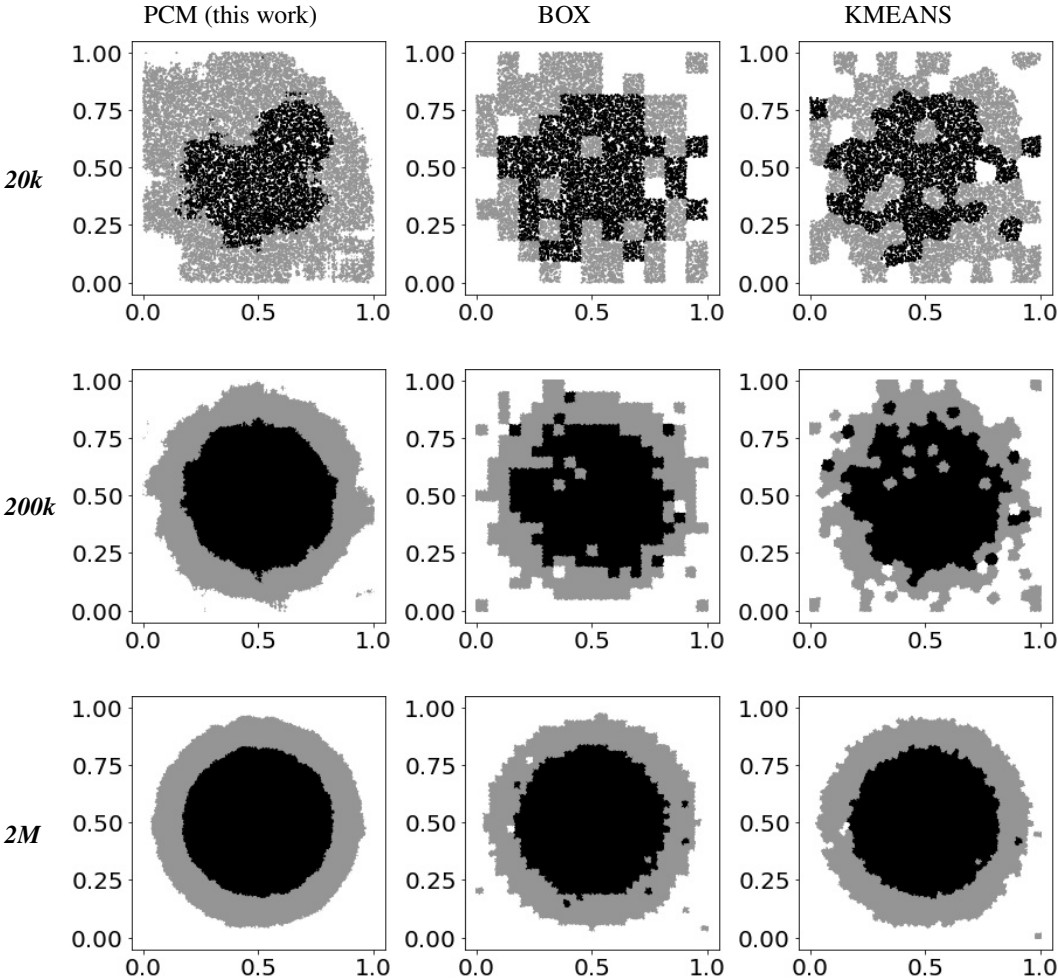

**Histograms.**

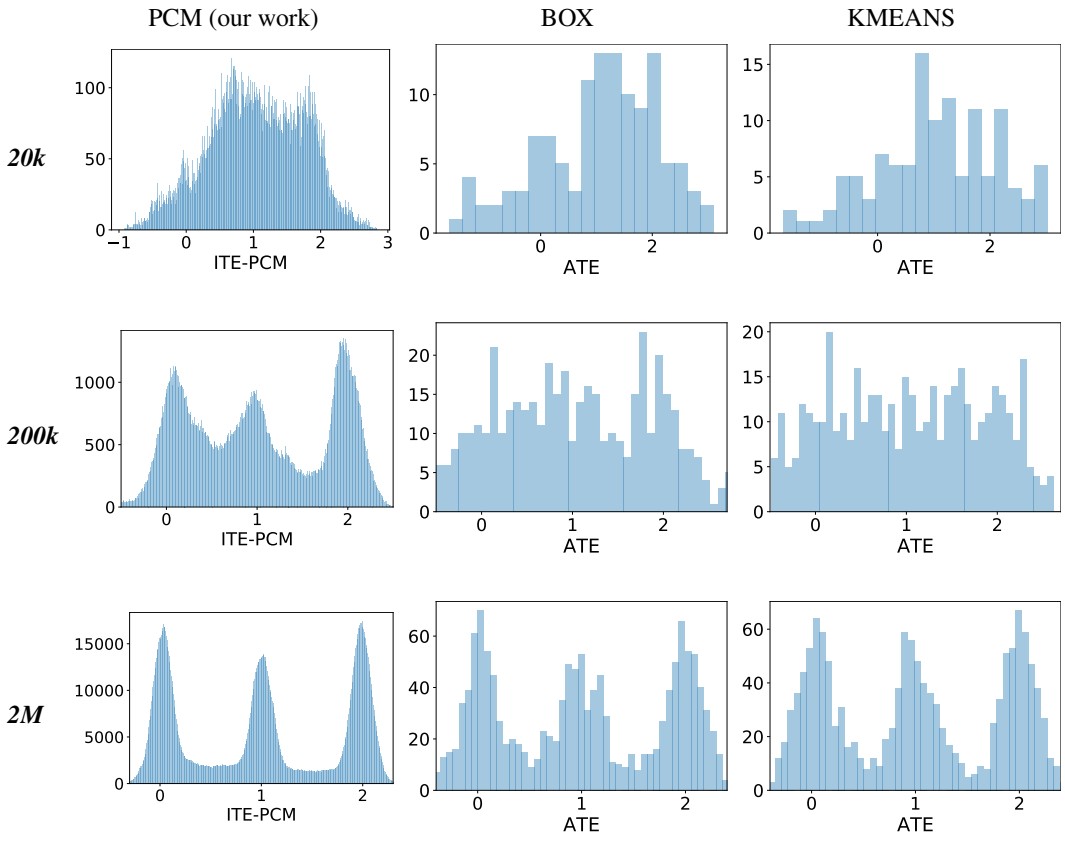

**Error Table.**

| $n$ | **PCM** (this work) | **BOX** | **KMEANS** |
|---|---|---|---|
| **20K** | **0.35±0.39** | $0.50 \pm 0.52$ | $0.54 \pm 0.50$ |
| **200k** | **0.109±0.22** | $0.17 \pm 0.35$ | $0.20 \pm 0.37$ |
| **2M** | **0.036±0.13** | $0.078 \pm 0.214$ | $0.065 \pm 0.20$ |

# D  CLUSTER HOMOGENEITY

To further show how practice reflects the theory, we plot average cluster homogeneity versus $n$. The cluster homogeneity is the fraction of points in a cluster that are from its majority level. Our entire methodology relies on the pre-clustering step producing a vast majority of homogeneous clusters. The rapid convergence to homogeneous clusters enables us to identify the correct subpopulations and the corresponding effects via pre-cluster and merge.

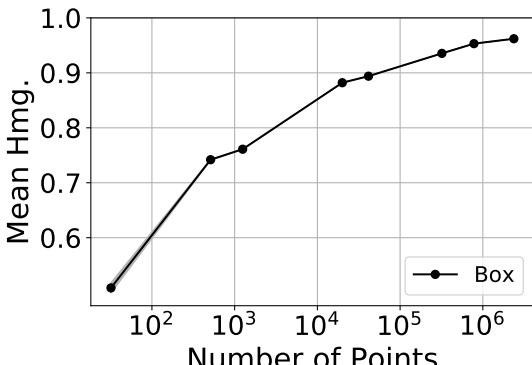

