# OpenReview forum: "Untangling Effect and Side Effect: Consistent Causal Inference in Non-Targeted Trials"
_ICLR.cc/2023/Conference — Submitted to ICLR 2023_

### Official Review · Reviewer_e3GP · 2022-10-25

**Confidence:** 3
**Correctness:** 3
**Technical Novelty And Significance:** 3
**Empirical Novelty And Significance:** 3
**Recommendation:** 6

**Clarity, Quality, Novelty And Reproducibility:**

Clarity

*The algorithm intuition is very clear

*Why is equation (2) about ATT instead of ATE? Relatedly, in the algorithm, isn’t the average of ITEs the ATE rather than the ATT? This was confusing

*The discussion of the difference between ATT and EFF used diction that is atypical in causal inference. A more typical discussion would say that under the stated identifying assumptions, ATT and EFF do not coincide. “rendering ATT useless” is a bit of a jarring expression.

*Some statements are too loose. “If unbiased counterfactual estimation is not possible, then any form of causal effect analysis is doomed.” Do the authors mean selection bias? Regularization bias? Either way, causal analysis is not doomed—there are entire literatures about how to correct for both biases. Please clarify which bias is being considered and fix the language.

*A figure would be helpful to visualize the different steps of the algorithm.

*I found this sentence to be confusing: “the counterfactual estimates use a predictor learned from the independent control population”. Is that the untreated subpopulation? Something else? Please clarify

Quality

*I crudely checked the results, but they seem rigorous

*I was surprised that the results did not appeal to smoothness of h or compactness of X_c. Is this not required for inverting h?

*A6 seems to be a strong assumption. Please discuss or formally sketch why this assumption is plausible. It would be nice to comment further on how the curse of dimensionality appears in A6. e^d N_{impure}<… seems to be the constraint. Is that correct?

*Overall, the strongest assumption seems to be that the treatment effect heterogeneity only depends on C, an integer, and the analyst knows this fact. This is formally stated in A2, but the introduction somehow didn’t convey this point forcefully enough.

Novelty

*The authors introduce the language of effect and side effect, but the overall goal is closely tied to a known problem that goes by several names: sorted group average treatment effect, classification analysis, and endogenous stratification analysis. I would like the authors to compare their problem statement with the problem statement of sorted group average treatment effect in Chernozhukov-Demirer-Duflo-Fernandez Val (2018). This reference also provides a thorough discussion of related works that are relevant for this submission and worth citing.

*While the problem is related to existing problems, the solution seems to me to be innovative and interesting.

**Strength And Weaknesses:**

See below

**Summary Of The Paper:**

The authors study the problem of estimating treatment effect heterogeneity when the subgroup indicators are unknown (e.g. sick versus healthy). In particular, the researcher knows that treatment effect heterogeneity only depends on the latent subgroup indicators, which are functions of observed covariates, but does not know how many subgroups there are and which subgroup each individual belongs to. The goal is to recover the correct number of subgroups, the effect for each subgroup, and to assign each individual to the correct subgroup. The authors propose an algorithm based on clustering and prove that it asymptotically achieves these three goals under seven assumptions.

**Summary Of The Review:**

I found the solution to be original and therefore significant. I will improve the score if the authors connect the problem statement with previous work (which will actually improve the significance).

---

> ### Comment · Reviewer_e3GP · 2022-11-18
> **This paper has potential**
>
> I was hoping the authors would further engage so I could improve the score. I think this paper is quite original and interesting, and with some improvements I would be willing to advocate for its acceptance

---

> > ### Author Response · Authors · 2022-11-18
> > **See response above.**
> >
> > We apologize for our delayed response.

---

> ### Author Response · Authors · 2022-11-18
> **The algorithm assumptions are mild and hence likely to hold in almost all practical settings.**
>
> We address the reviewers comments below.
> *Equation (2) is ATT because it is only over treated subjects. In a targeted trial this is ATE since only one effect-level is treated. In a non-targeted trial where multiple effect-levels get into the treatment group, this ATT is a mix of the ATEs from the different levels treated.
>
> *We apologize for using EFF. We will simplify to simply referring to effect levels. See previous comment about why the ATT does not reflect the CATE (conditioning on effect level). We agree with not using jarring statements and will refrain in a final version.
>
> *Again, we will refrain from language like doomed. We require that counterfactuals can be estimated, which means biases in the sampling can be corrected, and we use standard off-the-shelf techniques for this. What we meant to say is that if such biases cannot be corrected and counterfactuals estimated, then those biases will remain in the causal effect estimation.
>
> *We agree, we will give illustrative figures for each step in the algorithm, if space permits.
>
> *Regarding the counterfactual estimation, yes, the control population is the untreated population. Untreated is a better way to describe this population.
>
> *Yes, the assumptions are mild. We do not need smoothness, explicitly. We just need that the feature space is divided into levels according to the mapping h. We never need to actually compute h^(-1)(c), just that it exists. The extent to which we need smoothness is captured by the (mild) assumption A6. Essentialy it says that the boundary between levels cannot be so complicated (fractal) that it have a co-dimension close to the dimension of the space. So in a typical setting, for example in 2-dimensions, the boundary between effect-levels will be a one-dimensional curve. Such a boundary has measure 0. This means that the number of hypercubes needed to cover the boundary vanishes as a fraction of the number of hypercubes needed to cover the space, e.g. [0,1]^2. The curse of dimensionality appears indirectly because in higher dimension, the geometric size of the clusters is larger and so one only gets a coarse representation of the effect population h^(-1)(c). But asymptotically, we still extract the effect levels. This curse of dimensionality is pervasive in ML.
>
> *Yes, the crucial assumption is that there are a finite set of effect levels C. This is what allows us to extract those levels provably. This is an assumption for the provability of the algorithm. The algorithm can be run irrespective of this assumption and will provably extract a piecewise approximation to the full CATE function, in a non-parametric way. However, the proof of this is beyond the scope of this discussion and will involve additional much stronger assumptions as the reviewer pointed out, regarding the smoothness of the boundary. We will make it clearer in the intro that the theoretical results are conditioned on a finite number of effect-levels.*
>
> * We were unaware of the specific result in Chernozhukov-Demirer-Duflo-Fernandez Val (2018) but we did acknowledge that the problem we solve is a special case of HTE / CATE analysis. In the context of Chernozhukov-Demirer-Duflo-Fernandez Val (2018), which is an ML approach to CATE, the most related concept to our work is Sorted Group Average Treatment Effects (GATES), which attempts to get at a similar outcome of identifying the effect levels by *first* using the ML learned CATE to then infer the subpopulations. We go about it the opposite way. We use proximity based clusters to estimate local but noisy ATEs, and then MERGE to get the full subpopulations at the effect levels. We can prove consistency of our approach under mild assumptions. To prove consistency of GATES, one first needs consistency of the HTE method, which  Chernozhukov-Demirer-Duflo-Fernandez Val (2018) is very hard to do in the typical ML setting without very strong assumptions. This is natural because you are trying to solve a simpler problem by first solving a much harder problem. We are going directly for the simpler problem of the effect levels. We thank the reviewer for pointing us to this work.

---

> > ### Comment · Reviewer_e3GP · 2022-12-05
> > **Nice answers, but the paper is not updated**
> >
> > I appreciate the authors' thoughtful answers, which would lead me to improve the score. It seems the authors have not updated the paper, so I cannot verify whether and how these fixes have been made.
> >
> > If the fixes were made well, I would improve the score to 8 and advocate for acceptance. To my knowledge of the literature, the authors apply interesting and different analytical tools to this problem than previous work, which merits publication

---

### Official Review · Reviewer_KbyX · 2022-10-25

**Confidence:** 3
**Correctness:** 2
**Technical Novelty And Significance:** 3
**Empirical Novelty And Significance:** 2
**Recommendation:** 5

**Clarity, Quality, Novelty And Reproducibility:**

Paper is clear, seems like the PCM algorithm is relatively novel in this context (although the context of the entire fields it is difficult to discern given the lack of comprehensive related work section. No code to reproduce but should be easy.

**Strength And Weaknesses:**

Strengths
- Simple methods that works well.
- Proof of theorem 1 is methodical and clear.
- Robust to different clustering algorithms

Weaknesses
- Paper lacks refinement in exposition. For instance, I think a causal graph could have illustrative; defining groups (sick, health etc.) as a hidden confounder. I think cleaning this up and making this clearer would be helpful.
- The paper lacks a coherent related work section; would appreciate more related work for instance how this relates to estimating HTEs with hidden confounder.

Minor Comments
- Side effects doesn't seem like the appropriate nomenclature.
- ATT instead of ATE or CATE if conditional

Questions
- Experiments demonstrating performance in imbalanced dataset with respect to the groups would be useful
- Are there no baselines that are able to incorporate to incorporate using the HTE to identify groups?

**Summary Of The Paper:**

The paper proposes an algorithm (pre-cluster and merge) to better disambiguate heterogenous treatment effects in non-targeted clinical trials (where there is a hidden confounder of the patient of whether the patient was sick/healthy/other confounder variable-- with same observable patient properties/covariates otherwise.

**Summary Of The Review:**

I am leaning to reject, given the few weaknesses I've noted which I think are significant-- but would recommend accept should these be addressed.

---

> ### Author Response · Authors · 2022-11-18
> **Provable extraction of effect-levels**
>
> We agree that side-effect may not be conforming to the normal nomenclature. We will change it to simply effect-levels, and yes, we will add the causal graph.
>
> ATT is equal to the CATE when the treated population is from one effect-level, which occurs in a targeted trial. Since our focus is on non-targeted trials, these can be different. Our paper proposes a simple algorithm that provably extracts the CATE, conditioned on the effect level.
>
> All HTE methods are based on some form of regression methodology using the individual treatment effects (ITEs). In our experiments we did compare to one of the state-of-the art HTE-methods (X-Learner) to extract the groups, which resulted in inferior performance. The reason is that ITEs are noisy, which is why the pre-cluster and merge methodology is essential. While space restrictions prevent us from showing all results, we can include in the appendix results with imbalance. Our algorithm is robust to imbalance.

---

> > ### Comment · Reviewer_KbyX · 2022-11-29
> > **Response**
> >
> > - Nomenclature: Thank you. This wasn't addressed in the updated paper however.
> > - ATT vs CATE: is there a reference for this? This simply seems like CATE with a different conditional
> > - HTE imbalance results. Again should have been in updated paper.
> >
> > This paper has potential; I just don't think in its current state it's there yet.

---

### Official Review · Reviewer_7AgW · 2022-10-30

**Confidence:** 4
**Correctness:** 4
**Technical Novelty And Significance:** 1
**Empirical Novelty And Significance:** 1
**Recommendation:** 3

**Clarity, Quality, Novelty And Reproducibility:**

The writing could've been further polished, as many places use non-standard language, and others are somewhat confusing. It could be better to introduce the algorithm first before going through its theoretical properties. The proof can be differed to the appendix.

**Strength And Weaknesses:**

- A stronger justification of the practical significance of non-targeted trials could be greatly beneficial. In particular, although self-selection is indeed a well-known in e.g. natural experiments on digital platforms, it is a bit difficult for me to image self-selection problem would occur in clinical trials, where experimental units are recruited through a careful and rigorous process where doctors are usually involved. Therefore, perhaps a concrete clinical situation should be carefully elaborated, or relevant literature should be cited in the paper to substantiate the importance of this somewhat non-canonical situation. It is also important to show the situation of non-targeted trial is sufficiently common, so that any new method being developed for this situation is worth much attention.
- The algorithm in Section 2 looks like a algorithm for estimating heterogeneous treatment effect, but HTE is fundamentally different from the clinical sick/healthy subjects setting because usually it is pretty clear whether a patient is sick or healthy.
- The algorithm is a combination of cluster analysis and off-the-shelf estimation of individual ITE. Both methods are well known, so synthetic analysis could’ve taken less proportion because it is unlikely that the two methods would go wrong, especially the simulation setting is simplistic with dimension 2. On the other hand, real data analysis is necessary in order to evidence the value of the algorithm in application.

**Summary Of The Paper:**

The purpose of this paper is to provide statistical method to handle the so-called non-targeted trials in which one can’t control the selection of treatment units. Therefore, “some (often most) treated subjects are healthy (p.2)”, and this greatly confounds the estimation of ATT. This paper provides asymptotically consistent algorithm for accurately estimating the treatment effect, and demonstrate its value numerically with synthetic data.

**Summary Of The Review:**

The problem that this paper proposes to consider could have been better motivated by using a practical example, and the algorithm looks like a generic algorithm rather than one that centers at the proposed problem. A lack of a real data analysis also undermines the quality of this paper.

---

> ### Author Response · Authors · 2022-11-18
> **Our goal was to propose a simple algorithm which provably recovers the correct effect-levels under mild assumptions.**
>
> We agree that application on real-world application would improve the paper. However the goal of our paper was to propose a simple algorithm for which we can prove that it recovers the correct effect-levels under mild assumptions. We are unaware of methods that provably accomplish this. The purpose of the small experimental demonstration was to show the theory in action in comparison with state-of-the art existing packages which cannot perform as well under such mild assumptions.

---

> > ### Comment · Reviewer_7AgW · 2022-11-26
> > **I thank the authors for the answer, but unfortunately my questions remain largely unaddressed**
> >
> > My question has not been adequately addressed. First, I feel the proposed methodology isn't particularly novel, given that the proposed method looks like a straightforward combination of a standard cluster algorithm and off-the-shelf estimation of individual ITE. Second, I am not entirely convinced whether self-selection in clinical trials would actually happen in real world, given that subjects are often carefully selected by clinicians. My score remains unchanged because of these reasons.

---

### Decision · Program_Chairs · 2023-01-20

**Decision:**

Reject

**Justification For Why Not Higher Score:**

justification of the practical value of the proposed problem setting is needed

technical novelty needs further clarification, especially discussion on connection to related works is needed.

**Justification For Why Not Lower Score:**

n/a

**Metareview: Summary, Strengths And Weaknesses:**

This paper studies the estimation of treatment effects in the non-targeted trials in which one cannot control the selection of treatment units. An asymptotically consistent effect estimation algorithm is proposed and the experiments on simulated data are conducted to verify the effectiveness of the proposed algorithm.

While the paper introduces a new and interesting problem, there are several concerns on the technical quality and experiments. First, the new problem setting needs more justifications to show it is a nontrival setting and has real applications. Second, the technical novelty of the proposed method needs further clarifications. In particular, the relations to existing individual ITE methods need to be discussed. Finally, experiments on real world data would significantly strengthen the paper. Given these concerns, I would not recommend acceptance of this paper in its current form.  Although we think the paper is not ready for ICLR in this round, we believe that the paper would be a good one if the concerns can be well addressed.